# Tracking carrier protein motions with Raman spectroscopy

Samuel C. Epstein[1], Adam R. Huff[1], Emily S. Winesett[1], Casey H. Londergan [1] & Louise K. Charkoudian [1]

Engineering microbial biosynthetic pathways represents a compelling route to gain access to expanded chemical diversity. Carrier proteins (CPs) play a central role in biosynthesis, but the fast motions of CPs make their conformational dynamics difficult to capture using traditional spectroscopic approaches. Here we present a low-resource method to directly reveal carrier protein-substrate interactions. Chemoenzymatic loading of commercially available, alkyne-containing substrates onto CPs enables rapid visualization of the molecular cargo's local environment using Raman spectroscopy. This method could clarify the foundations of the chain sequestration mechanism, facilitate the rapid characterization of CPs, and enable visualization of the vectoral processing of natural products both in vitro and in vivo.

[1] Department of Chemistry, Haverford College, Haverford, PA 19041-1391, USA. Correspondence and requests for materials should be addressed to C.H.L. (email: clonderg@haverfored.edu) or to L.K.C. (email: lcharkou@haverford.edu)

Microbial combinatorial biosynthesis represents a potentially powerful route to gain access to expanded chemical diversity from renewable resources. Its success hinges on understanding how proteins within a synthase communicate. Current mix-and-match approaches often fail due to incompatibility between carrier proteins (CPs) and other enzymes, although the reason why is not entirely clear[1]. CPs are dynamic proteins that play the most central role during the biosynthesis of pharmaceutically important classes of molecules, such as fatty acids, polyketides, and non-ribosomal peptides (Supplementary Fig. 1). These small proteins interact with virtually all other proteins within the synthase, and they tether a variety of molecular building blocks and intermediates during natural product biosynthesis[2].

While visualizing CP conformational changes and interactions with other species is essential for creating functional hybrid synthases, directly capturing transient interactions and the full ensemble of CP conformations remains a challenge for at least two reasons[3,4]. First, catalytically relevant CP movements are thought to occur on the micro- to pico-second (μs to ps) timescale[5], and thus neither nuclear magnetic resonance spectroscopy (NMR) nor X-ray crystallography can provide a direct picture of how the conformational distributions of CPs change during catalysis. Second, these traditional protein structure methods are time-, resource-, and sample-intensive.

A conformationally mobile and important feature of CPs is the 18 Å 4′-phosphopantetheine (Ppant) arm, which is attached post-translationally to a conserved serine residue typically at the N-terminal end of CP helix II[6]. The Ppant arm covalently tethers all building blocks and intermediates as thioesters, and its flexibility enables the CP to sequester specific molecular cargoes within its hydrophobic cavity[7]. This chain sequestration is believed to protect the growing metabolite from undesired chemical reactions with cytoplasmic components and/or drive other overall conformational changes that can enhance the specificity of a CP for a particular enzymatic partner[8,9]. Sequestration has been primarily observed in CPs from type II systems with discrete enzymes that act iteratively, where protection and transportation of the intermediate to the appropriate enzymatic partner at the programmed stage in biosynthesis are of the utmost importance in maintaining chemical fidelity[10]. In contrast, modular type I synthases with covalently linked catalytic domains do not typically exhibit substrate sequestration[11]. The precise molecular underpinnings of chain sequestration remain unknown but are thought to involve the interplay of at least three factors: CP sequence, substrate length, and substrate polarity[12].

Here we report a facile and low-resource method to visualize the full ensemble of CP-substrate interactions using site-specific vibrational spectroscopy (Fig. 1). In brief, molecular substrate-mimics with terminal alkyne probes are installed onto the CP via the ligase-catalyzed addition of commercially available carboxylic acids onto the Ppant arm. The alkyne C≡C stretching band is then used to report on changes in the probe environment, which can differentiate with picosecond time resolution between the non-sequestered aqueous state (lower frequency) and the sequestered hydrophobic environment (higher frequency). The modification of a native substrate to include a terminal alkyne is expected to only minimally perturb the natural system because it does not alter the overall length, volume, or hydrophobicity of the molecular cargo. The alkyne C≡C stretching band is a strong, narrow, and unique signal in the transparent region of the Raman spectrum (close to 2100 cm$^{-1}$) that does not overlap with other solvent or biomolecular signals from the untagged CP or other proteins.

## Results

**Characterization of chain sequestration behavior.** For proof-of-concept experiments, we collected data from three acyl carrier proteins (ACPs) for which chain sequestration information was previously reported via NMR and molecular dynamics (MD) simulations: the *E. coli* type II fatty acid synthase (FAS; EcACP), *Streptomyces coelicolor* type II actinorhodin polyketide synthase (PKS, Act ACP), and the mammalian rat type I FAS (Rat ACP). For EcACP, MD simulations suggested that an octanoyl acyl chain is the ideal length for complete sequestration of the molecular cargo inside the ACP hydrophobic core[13]. Shorter acyl chains were proposed to be highly mobile and less sequestered since the ACP cavity is too large to stabilize the short substrates; and larger acyl chains are sequestered only at the terminal end of the chain[14]. Previous NMR analysis of Rat ACP in various acylated states suggested that Rat ACP does not sequester any hydrophobic acyl-intermediates due to bulky hydrophobic

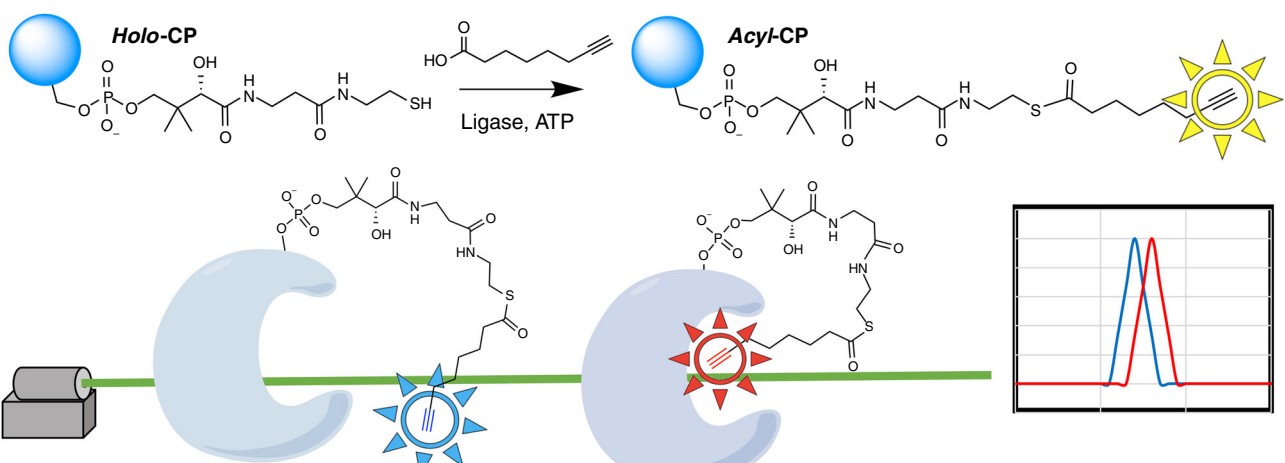

**Fig. 1** Workflow for determining CP-substrate interactions via Raman spectroscopy. An alkyne-labeled fatty acid (of selected length) is ligated to the terminal thiol of the Ppant arm via the promiscuous ligase AasS (top). The probe-labeled molecular cargo serves as a reporter of whether a substrate is sequestered into the hydrophobic cavity of the CP through changes in the Raman scattering spectrum. The C≡C frequency reports on the solvation environment (lower frequency when the probe is in an aqueous environment, or higher frequency when the probe is in the protein's hydrophobic cavity) and the line shape reports on the ps-resolved range of conformations

residues that line the interior pocket[11]. Rat ACP should be viewed as an analog of the EcACP with an inhibited sequestration capability because Rat ACP has been shown to at least partially substitute for the EcACP in vitro, functionally interacting with the acyltransferase, ketosynthase, and reductase domains from the *E. coli* FAS[15]. Taken together, the EcACP and Rat ACP systems present an ideal juxtaposition for preliminary experiments. NMR studies of Act ACP reveal interesting and distinct behavior: butyryl-, hexanoyl-, and octanoyl- acyl chains bind within the hydrophobic cavity, but the substrates are situated perpendicular to their traditional orientation, possibly due to the large size of the cavity (Supplementary Data 1)[12].

His-tagged ACPs were expressed and purified from *E. coli* BAP1 competent cells[16]. If necessary, ACPs were completely phosphopantethienylated using the R4-4 Sfp transferase from *B. subtilis* (Sfp)[17]. The *V. harveyi* acyl-ACP synthetase (AasS) was used to acylate the alkyne-containing carboxylic acid to the terminal thiol of the Ppant arm[18]. 7-octynoic acid (a mimic for an eight-carbon $C_8$ substrate) was loaded onto both EcACP and Rat ACP, producing distinct Raman spectra in the region of the alkyne probe signal (Fig. 2a). EcACP, expected to sequester the $C_8$ cargo based on literature precedent[13], exhibited a higher frequency than that of the $C_8$ probe in buffered aqueous solution, consistent with chain sequestration. Conversely, the Rat ACP probe frequency and lineshape were nearly identical to that of the solvated probe, indicating that the same chain attached to Rat ACP was not sequestered. The probe Raman spectrum of Act ACP loaded with 7-octynoic acid is broader and covers frequencies associated with both hydrophobic and aqueous environments, in agreement with NMR data for octanoyl Act ACP that suggested the $C_8$ substrate was only partially sequestered (Supplementary Data 1)[19]. Taken together, these results from already-characterized ACPs validate our Raman probe-based approach to visualize CP chain sequestration. The spectra in Fig. 2a highlight how the C≡C frequency reports sequestration: the line shape directly reports the complete distribution of environments experienced by the probe, thus providing direct information about the heterogenous nature of substrate sequestration with sufficient temporal resolution. The intrinsic timescale of Raman spectroscopy for this vibrational probe is about 10 ps (see Supplementary Fig. 2 for more detail); any configurations that interconvert more slowly can be distinguished in the spectral lineshape.

Next, we used this technique to directly evaluate the role of the acyl chain length in chain sequestration. EcACP was acylated with 4-pentynoic acid (a $C_5$ substrate) and 12-tridecynoic acid (a $C_{13}$ substrate). Previous crystallography and NMR studies of acyl EcACPs suggested that hexanoyl, heptanoyl, and decanoyl chains were fully sequestered, while the precise range of butyryl-bound substrate environments remained unclear[7,20]. The Raman spectrum of the $C_5$ probe on EcACP indicates that this probe is not sequestered, whereas the $C_{13}$ probe displays a similar spectrum to that of the sequestered $C_8$ probe (Fig. 2b). These data suggest that the two longer probe-labeled chains are sequestered; this provides insight into how EcACP interacts with longer chain lengths, such as $C_{13}$, which has not been the subject of previous work. These results, from different-length substrates on the same CP, point towards future application of our Raman-based technique to explore how substrate sequestration changes throughout the entire substrate elongation process.

This method provides a quick, low-cost, and effective means to analyze CP-substrate interactions that does not depend on structural rendering of the entire protein through more labor-intensive methods. All steps in the process are amenable to high-throughput approaches, which will facilitate rapid characterization of CPs for which conventional structural data are not available. To examine chain sequestration in these CPs, we ligated the $C_8$-alkyne probe onto the Ppant arm of two previously uncharacterized type II PKS ACPs: arimetamycin (Arm ACP) and benastatin (Ben ACP), as well as the ACP from the spore pigment biosynthetic gene cluster, WhiE ACP (Fig. 2c). The Arm ACP spectrum exhibits a higher C≡C frequency than the $C_8$ probe in buffered solution. Like Rat ACP, the Ben ACP and WhiE ACP spectra produced nearly identical signals to those of the aqueous $C_8$ probe. The spectrum of the WhiE ACP loaded with the $C_5$ probe also indicated a non-sequestrated state (Supplementary Fig. 3). Alongside data from the $C_8$ probe on Act ACP, these results support the hypothesis that a general feature of type II PKS ACPs is a hydrophobic cavity too large to fully stabilize shorter and/or less polar acyl chain probes relative to the native substrate[12]. Coupling the Raman probe technique described here with the synthesis of more sophisticated substrate-intermediates, and/or site-directed mutagenesis of tagged ACPs, will reveal further details of the exact molecular

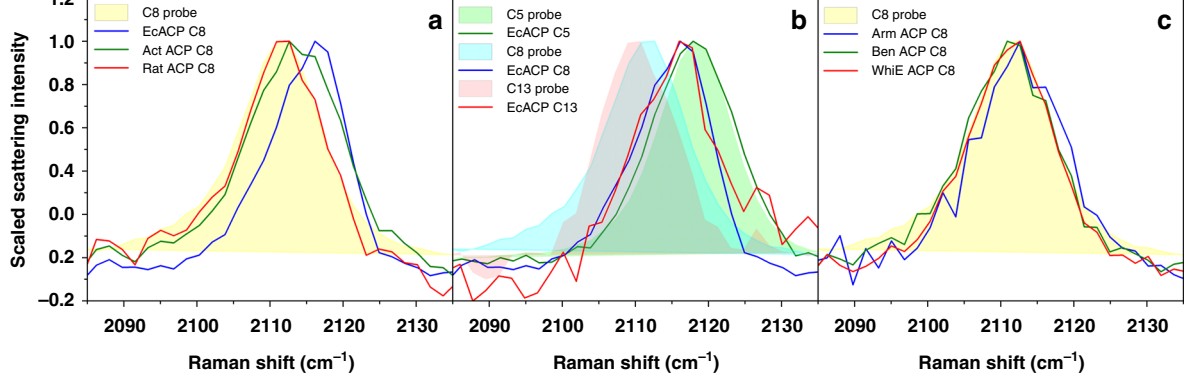

**Fig. 2** Raman scattering of C≡C modified substrates reports on CP chain sequestration. **a** Raman spectra for EcACP (blue), Act ACP (green), and Rat ACP (red), each loaded with the $C_8$ probe, provide information about the local environment of the C≡C probe consistent with literature precedent that a $C_8$ substrate chain is sequestered by EcACP, not sequestered by Rat ACP, and partially sequestered by Act ACP. **b** Raman spectra for $C_5$ (green), $C_8$ (blue), and $C_{13}$ (red) probes on EcACP show how sequestration depends on the chain length. **c** Spectra for $C_8$ probe (yellow) on Arm ACP (blue), Ben ACP (green), and WhiE ACP (red) provide chain sequestration information about previously uncharacterized ACPs. In all cases, line spectra represent data collected for probes on ACPs, and shaded bands represent data for free alkyne-labeled carboxylic acids in buffered aqueous solution. In all cases, a shift to higher frequency indicates that the alkyne probe enters a more hydrophobic environment as it becomes sequestered inside the hydrophobic pocket of an ACP. (Source data are provided as a Source Data file.)

interactions that govern chain sequestration in both ACPs and CPs from other biosynthetic pathways.

## Discussion

The ubiquitous yet heterogeneous (across substrate lengths and proteins from different pathways) role of CP-substrate interactions is central in natural product biosynthesis, yet conventional structural methods cannot directly capture these events. The site-specific vibrational approach implemented here represents a relatively simple and broadly applicable method that will enable the rapid elucidation of dynamic structures across diverse CP-substrate interactions. The optical equipment used here (see Methods) is an ordinary continuous-wave, dispersive Raman spectrometer that does not supply any optical enhancement (i.e. UV-resonance or stimulated scattering) of the Raman signal, so signals like those we report are quite easy to access using relatively unspecialized equipment. While the interpretations that we present of the alkyne frequencies and lineshapes are based on empirical comparisons, recent work has demonstrated that MD simulations coupled with effective fragment potential-based calculations can be used to quantitatively simulate vibrational probe lineshapes[21,22]. Our current computational work focuses on the extension of such calculations to alkyne probes and the simulation of the spectra observed here; these efforts should enable a more directly quantitative and physical interpretation of the Raman data in Fig. 2 and from other CPs of future interest.

The alkyne probes introduced onto the CPs in this study could also serve as bioorthogonally reactive substrates capable of being processed through chain elongation, chain transfer, and tailoring events while simultaneously reporting on changes in the local substrate environment. With strong and growing evidence that CP chain sequestration and flipping (the movement of a CP-bound substrate from inside the hydrophobic core of the CP into the active site of a partner enzyme) is centrally linked to functional channeling of biosynthetic intermediates[8], this approach can be applied broadly to fill a central and unmet need in understanding the molecular details of those biosynthetic pathways across many species and synthases.

It is also anticipated that this technique will be used to elucidate the unconventional behaviors of CPs and can be applied in cases where, for example, the substrate is tucked against a non-polar patch on the surface of the CP domain[23]. Probe-labeled substrates, including those containing alkynes further up the chain and those with more complicated oxidation and substitution patterns, can also be utilized to provide more in-depth insight into the nature of CP-substrate interactions. Additionally, the alkyne probes in this work could also bridge the gap between in vitro and in vivo studies of biosynthetic events, as alkyne-labeled species can be imaged in vivo by stimulated Raman microscopy (sometimes simultaneously with other fluorescently labeled species, which could enable novel co-localization studies of direct relevance to biosynthesis)[24–26]. In a more general sense, data from the technique reported here could help to design hybrid natural product synthases capable of accessing novel chemical diversity.

## Methods

**Protein expression and purification**. BAP1 competent cells[16], which feature a T7 RNA polymerase, were used to transform the respective plasmids for expression of EcACP, Act ACP, Rat ACP, Arm ACP, Ben ACP, and WhiE ACP (all featuring kanamycin resistance, except for Act ACP which featured carbenicillin resistance). The EcACP plasmid pTL14[27] (N- and C-terminal His$_6$-tagged) was provided by the Khosla Lab at Stanford University. The Act ACP plasmid (pMC002067; carbenicillin resistant) was provided by the Chang Lab at University of California, Berkeley. For the remaining ACPs, plasmids were designed via the following protocol: ACP sequences were purchased from integrated DNA technologies (IDT) as gBlock DNA fragments, and 100 ng of the DNA (dissolved in water) was digested (30 uL reaction) using 1 µL NdeI and 1 µL EcoRI (or BamHI) with 10X

CutSmart buffer (New England Biolabs). The mixture was then incubated at 37 °C (12 h). A QIAprep Miniprep kit (Qiagen) was used for DNA purification. To precipitate the DNA, the mixture was treated with 2.5 µL of 3 M sodium acetate, 2 µL glycogen, and 200 µL ethanol, and then stored at 20 °C (12 h). The supernatant was washed (200 µL of 70% ethanol), dried, and then suspended in 1X DNA dilution buffer. For ligation (using T4 DNA Rapid Ligation Kit, Roche), the digested DNA insert was added to 100 ng of gel-purified (Zymoclean) pET28a vector (featuring an $N$-terminal His$_6$-tag) digested with NdeI/EcoRI and treated with calf-intestine alkaline phosphatase (CIP), T4 ligase, and 1X Dilution buffer. After incubation at room temperature (30 min), 10 µL of the ligation product mixture was transformed into chemically competent DH5α cells and plated on LB agar plates (50 µg/mL kanamycin). The plasmids for Arm ACP, Ben ACP, and WhiE ACP were prepared by these means for a previous study[28]. The plasmid for Rat ACP was prepared for this study (see Supplementary Fig. 4 for sequence of DNA insert). All plasmids are available from the authors upon request.

A single colony was selected for the growth of seed cultures overnight at 37 °C in 10 mL of LB media (50 µg/mL of kanamycin or 100 µg/mL of carbenicillin). Seed cultures were then added to 0.75 L production cultures (50 µg/mL of kanamycin or 100 µg/mL of carbenicillin) and were grown at 37 °C until the OD$_{600}$ = 0.6. After sufficient culture growth, cells were induced with 188 µL of 1 M IPTG. The induced cultures were incubated at 18 °C overnight, while shaking. Following the incubation period, cells were harvested by centrifugation (5,000 RPM, 20 min), and the cell pellet was stored at −80 °C. Next, the cells were thawed on ice, resuspended in lysis buffer (50 mM sodium phosphate pH 7.6, 300 mM NaCl, 10 mM imidazole), and sonicated (cells on ice at 4 °C, 10 × 30 s pulses with 30 s rest in between, 40% power). Cell debris was removed by centrifugation (13,000 RPM, 45 min). Nickel-NTA agarose slurry was equilibrated into lysis buffer by repeated centrifugation and decanting (3 times), before adding to the protein-containing supernatant (4 mL slurry/L starting culture). The protein-nickel resin mixture was left to mix for 1 h at 4 °C. The mixture was allowed to settle for 15 min and the supernatant was decanted carefully with a serological pipette. The resin was loaded onto a fritted column, allowing the flowthrough to be collected. The resin was then washed with 100 mL of wash buffer (50 mM sodium phosphate pH 7.6, 300 mM NaCl, and 30 mM imidazole). Finally, the desired proteins were eluted with 10 mL of elution buffer (50 mM phosphate, pH 7.6, 100 mM NaCl, and 150 mM imidazole). The concentrations of the protein containing elution fractions were confirmed via Nanodrop measurements. Desired fractions were pooled and buffer exchanged using a 3.5-kDa MWCO 0.5 mL–3 mL Thermo Scientific™ Slide-A-Lyzer™ Dialysis Cassette into 50 mM phosphate buffer pH 7.6. Proteins were aliquoted and flash frozen for storage at −80 °C.

**Complete 4′-phosphopantetheinylation of ACPs**. Liquid chromatography mass spectrometry (LC-MS) analysis of ACPs (see below) revealed that some ACPs were produced as a mixture of the *apo* and *holo* forms and thus a 4′-phosphopantetheinyl transferase was used to push these ACP samples completely to the *holo* form. Coenzyme A (1.5 mM) was added to a solution of ACP (0.95–1.0 mM) with Sfp R4-4 (2 µM)[17], magnesium chloride (10 mM) and DTT (2.5 mM), in 50 mM sodium phosphate pH 7.6 (0.8–1 mL total volume). The solution was incubated for 24 h at room temperature. All ACPs were purified into 50 mM phosphate buffer pH 7.6 using an AKTA FPLC equipped with a HiPrep 26/10 de-salting column. EcACP was concentrated to 1 mM using a 3.5-kDa MWCO Centricon centrifugal filter. LC-MS was used to confirm all ACPs were completely in the *holo* form before substrate loading.

**Chemoenzymatic attachment of probe-containing substrates**. 4-pentynoic acid (C5), 7-octynoic acid (C8), and 12-tridecynoic acid (C13) were all commercially acquired (C5:Enamine, C8:Enamine, C13:Mcule). NMR data were acquired to verify the purity of each probe (Supplementary Figs. 5–7). Carboxylic acids were loaded onto the ACPs using the *Vibrio harvei* Acyl-acyl carrier protein synthetase (AasS). AasS has been shown to be a promiscuous ligase, capable of loading various fatty acids onto the terminal thiol of the ACP's Ppant arm[18]. The reaction was completed on a 1-mL scale, consisting of the ACP (275 µM, stock in 50 mM sodium phosphate buffer pH 7.6), dithiothreitol (2.5 mM), magnesium chloride (23 mM), ATP (18 mM, stock adjusted to pH 7.6), AasS (0.8 µM, stock in Tris buffer pH 7.6), and the desired carboxylic acid (4.6 mM, stock in isopropanol). The reaction was prepared in a glass vial, as plastic tubes have previously been shown to contain competing carboxylic acids that could be loaded onto the ACP in place of the desired substrates. The reaction mixture was left shaking at 100 RPM for 16 h at 37 °C. Samples were spun in a centrifuge at 13,000 RPM for 5 min to pellet precipitation. Supernatant was loaded onto a Sephadex G-25 in PD-10 Desalting Column to separate the protein from salts and remaining unloaded substrate. A ThermoScientific Nanodrop 2000c spectrophotometer was used to determine protein concentration and purity. Fractions featuring characteristic protein peaks and lacking a 260-nm peak (characteristic of the unloaded carboxylic acid) were selected. Chosen fractions were pooled and concentrated using a 3.5-kDa MWCO Centricon centrifugal filter. The concentration used for visualization via Raman spectroscopy ranged from 1 to 3 mM.

**Verification of substrate loading onto ACPs**. ACPs (20 µL of a 0.1-mg/mL solution in 50 mM sodium phosphate buffer, pH 7.6) were analyzed by LC-MS

(AgilentG6125BW) to confirm the success of the loading reaction (Supplementary Figs. 8–23). A Waters XBridge Protein BEH C4 Column (300A, 3.5 μm, 2.1 mm × 50 mm) heated to 45 °C was used for analysis by ESI-MS in the positive mode. The following gradient was used (solvent A = water + 0.1% formic acid; solvent B = acetonitrile + 0.1% formic acid): 0–1 min 95% A, 3.1 min 5% A, 4.52 min 5% A, 4.92–9 min 95% A. Data were deconvoluted using ESIprot[29], a free, online software, and the observed MW was compared to the calculated MW for *holo*- and *acyl*-ACPs. All ACPs eluted from the column at 4.4 min.

To determine if excess unloaded probe remaining in solution could be detected by LC-MS, free carboxylic acids were added to probe-loaded ACPs at target concentrations (2.5 mM *acyl*-Act ACP, 250-μM-free probe) to ensure that free probe did not remain in the acyl-ACP samples after purification (Supplementary Figs. 24–26). SDS-PAGE (Supplementary Fig. 27) and urea PAGE (Supplementary Fig. 28) were also used to distinguish ACPs with different size and conformation. SDS PAGE analyses were performed using 12% acrylamide (1 mm) in Tris-HCl gels. 5% (v/v) 2-mercaptoethanol was added to the sample loading dye. All samples were boiled at 95 °C and the gel was run at 120 V for about 80 min using premade running buffer (0.25 M glycine, 0.375 M Tris-HCl pH 8.8, 0.1% (w/v) SDS, Jule Biotechnologies). Precision Plus Protein Standard (BioRad) was used as the protein standard. Urea PAGE analyses were performed using 20% urea, 12% acrylamide (1 mm) in Tris-HCl gels. The gel was run at 185 V for about 100 min using premade running buffer (0.25 M glycine, 0.375 M Tris-HCl pH 8.8). All gels were washed in ddH₂O, stained at room temperature for 20–30 min using Coomassie SafeStain (Thermo Fisher Scientific), and finally destained overnight at room temperature in ddH₂O.

**Characterizing ACP secondary structure after probe loading**. Circular dichroism (CD) spectroscopy was performed using an Aviv 410 spectrophotometer (Aviv Biomedical, NJ). The secondary structures of *holo* and modified ACPs were determined by obtaining CD spectra at far-UV (260–180 nm) in a 0.1 mm path length cuvette (Hellma Analytics). All data were collected at 25 °C using a 1-nm bandwidth, a step resolution of 0.5 nm, and a 3-s averaging time. The baseline was corrected against the storage buffer, and ACPs concentrations were ~0.5 mg/ml. The corrected spectra were smoothed using a manual smoothing function implemented in the instrument software, using a window width of 11 data points, degree 2. Smoothed data were plotted in Origin (v.8.6.0). See Supplementary Figs. 29–30.

**Raman spectroscopy**. All Raman spectra were collected using a home-built CW-Raman spectrometer. A 532-nm DPSS CW laser (Cobolt, Inc.) attenuated to 80 mW incident power was focused vertically through a 1-mM diameter glass capillary filled with 1–5 μL of sample. Scattered light was collected at 90° to the incoming excitation using a Nikon f/1.2 camera lens and then focused into the slit of a PI-Acton SpectraPro 500 mm single monochromator (with a 600 grooves/mm grating blazed at 500 nm) and collected on a PI-Acton Spec10/100 liquid-nitrogen cooled CCD camera. Rayleigh scattering was rejected using a > 532 nm long-pass filter (Edmund Optics). Spectra were collected in exposures of 1 min for up to 2 h total accumulation time. All ACP samples were 1–3 mM concentration.

Data analysis was completed in Origin 8. The raw data was imported and was analyzed within a large region around the desired peak to establish a baseline (2000 cm⁻¹ to 2300 cm⁻¹). A smaller region containing the peak (2100 cm⁻¹ to 2130 cm⁻¹) was cut out to introduce a hole where the peak of interest is. The hole-containing baseline region was fit to a seventh-degree polynomial, and this polynomial was subtracted from the peak-containing baseline region data. The alkyne probe signal was then scaled to set the maximum point of the peak of interest as 1.0 and all other points a fraction relative to the maximum. The mode (from inspection), mean (calculated between 2100–2130 cm⁻¹), and FWHM (by inspection) are reported for each spectrum (Supplementary Table 1).

**Reporting summary**. Further information on research design is available in the Nature Research Reporting Summary linked to this article.

## Data availability

The data underlying the findings of this study are available from Open Science Framework (https://doi.org/10.17605/OSF.IO/RKD4E) and from the authors upon reasonable request. The raw data underlying Fig. 2 as well as Supplementary Figs. 3 and 27–30 are provided in a Source Data file.

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

## Acknowledgements

We thank Ashley C. Sisto and Dr. Bashkim Kokona (Haverford College) for their technical support. We are grateful to Dr. Joris Beld (Drexel University), Dr. Kara Jaremko (Hofstra University), and Dr. Benjamin Thuronyi (Williams College) for helpful discussions. We acknowledge NIH R15GM120704 (L.K.C.), NSF CHE- 1800080 (C.H.L.), a Henry Dreyfus Teacher Scholar Award (C.H.L.), and the Barry Goldwater Scholarship (S.C.E.) for funding. The content is solely the responsibility of the authors and does not necessarily represent the views of the NIH or NSF.

## Author contributions

L.K.C. and C.H.L. designed the study; S.C.E., E.S.W., and A.R.H. performed research; S.C.E., E.S.W., A.R.H., C.H.L., and L.K.C. analyzed data; S.C.E., C.H.L., and L.K.C. wrote the manuscript.

## Additional information

**Competing interests:** The authors declare no competing interests.

