## [Peer Review File · Nature Communications]

Reviewers' comments:

Reviewer #1 (Remarks to the Author):

The authors report that Raman spectroscopy of alkyne probes can be useful to detect chain sequestration of carrier proteins. The simple experiments using the combinations of various carrier proteins and substrates presented the different behaviors of Raman spectra of alkyne at the terminal of substrates, which indicate the change of the local environment around the substrate. The procedure of the experiments and material analysis are adequately performed. I agree that the technique would be beneficial to detect carrier protein - substrate interactions. However, it is necessary to confirm that the changes in Raman spectra induced by the sequestration. This was supported by qualitative considerations based on the results of the structural analysis. It would be convincing if the authors could compare Raman spectra with and without inhibiting or disturbing the sequestration capability. Trying QM/MM is also interesting to simulate the Raman shift of alkyne under sequestration.

Reviewer #2 (Remarks to the Author):

This is a valuable contribution to the acyl carrier protein field. The sample preparations and quality control are exemplary. The data are solid.

However, the authors state that these are facile and quick measurements. That seems true if one has the proper Raman spectrometer. The authors used a home-built instrument. Are there commercial instruments of similar abilities? If so, how likely are they to be accessible to biochemists? Most Raman spectrometers I see are in material sciences labs. The authors need to address this aspect.

Reviewer #3 (Remarks to the Author):

In this short, well written report, the authors introduce a new method for assessing the solvent exposure of acyl substrate mimics that have been covalently attached to the 4'-phosphopantetheine (Ppant) arms of acyl carrier protein (ACP) domains. The approach constitutes an ingenious extension of previous techniques devised by the corresponding author, who first followed changes in the ^{19}F NMR signal of a trifluoromethyl group appended to the Ppant prosthetic group and then monitored changes in the vibrational signature of a similarly attached thiocyanate probe using infrared absorption spectroscopy. The current advance is to use Raman spectroscopy to detect changes in the signature of a carbon-carbon triple bond incorporated into Ppant-attached acyl substrate mimics of various lengths. The method can indicate whether the alkyne group experiences a non-polar or an aqueous environment in the absence of more detailed but time-consuming structural studies by X-ray crystallography or NMR spectroscopy. It will therefore be useful for testing the idea that type II fatty acid synthase (FAS) and polyketide synthase ACP domains sequester their substrate into a hydrophobic cavity, while type I ACPs leave their substrates solvent-exposed.

Overall, in my opinion the article is of wide interest and is worthy of publication in this journal. However, I also feel that the authors need to address the following points:

(1) If the alkyne substrate-mimic appears to remain solvent-exposed when attached to an ACP domain, an alternative explanation is that the protein might not be properly folded. The authors do not rule out this possibility in their report. From one of the corresponding author's recent publications (Rivas et al, 2018), it seems that circular dichroism data has been collected for all of the domains studied except for the rat FAS ACP. Such information indicates that at least the expected alpha-helical secondary structure is present, so it should be referred to here, perhaps as a final validation step in the "Protein expression and purification" section of the Methods. Confirmation that the rat FAS ACP is structured should also be presented. The phosphopantetheine transferase Sfp is promiscuous enough to modify short unstructured peptides that possess the correct recognition sequence (e.g. see Yin et al (2005) PNAS 102, 15815-15820); successful modification by this enzyme (as observed here for all cases) is therefore not sufficient to prove

that an ACP domain possesses native tertiary structure.

(2) In addition to full burial and full exposure of the acyl chain, a third option is that substrate-mimics may nestle against a non-polar patch on the surface of the ACP domain (e.g. see Vance et al (2016) *Biochem J* 473, 1097-1110). It is not obvious how this mode of interaction would affect the vibrational frequency of the alkyne group, so it might be appropriate for the authors to include it in their suggestions for future work.

(3) All of the substrate-mimics tested in the report possess chain-terminating alkyne groups. It would be interesting to learn how the exposure of this group changes if it is moved further up the chain. Perhaps this could also be included in the list of future applications.

(4) In line 143, the WhiE ACP is not introduced adequately. It also appears that this construct not fully described elsewhere in the text, or in the prior publication Rivas et al (2018). I imagine that the authors intend this to indicate the product of the WhiE gene from the grey spore pigment biosynthetic cluster in *Steptomyces* strain. The source of this construct should be described in the text more fully.

(5) In line 111, the claim to have achieved "unprecedented temporal resolution" requires some moderation. The authors explain convincingly why their method ought to be capable of achieving temporal resolution, but the experiments presented here do not show this

(6) In line 42, the modifiable serine residue is described as being "at the end of CP helix II". It would be more instructive to change this statement to "at the N-terminal end of CP helix II"

Reviewer #1 (Remarks to the Author):

The authors report that Raman spectroscopy of alkyne probes can be useful to detect chain sequestration of carrier proteins. The simple experiments using the combinations of various carrier proteins and substrates presented the different behaviors of Raman spectra of alkyne at the terminal of substrates, which indicate the change of the local environment around the substrate. The procedure of the experiments and material analysis are adequately performed. I agree that the technique would be beneficial to detect carrier protein - substrate interactions. We are pleased to hear that the Reviewer feels that the technique we presented will be beneficial to the field!

*However, it is necessary to confirm that the changes in Raman spectra induced by the sequestration. This was supported by qualitative considerations based on the results of the structural analysis. It would be convincing if the authors could compare Raman spectra with and without inhibiting or disturbing the sequestration capability. We thank the Reviewer for this helpful comment, which made us realize that we needed to more clearly articulate the goals of using the Rat ACP. The Type II ACP from the *E. coli* FAS, loaded with a C8 substrate, was selected to represent the most idealized condition of an ACP fully sequestering its substrate, based on molecular dynamics simulations performed by other groups. Likewise, the type I ACP from the rat FAS was selected as a foil to the *E. coli* ACP due to characterization by NMR that it could not sequester any acyl substrates. Considering that Type I and Type II FAS ACPs are highly related and the Rat ACP has been shown to at least partially substitute for the *E. coli* ACP, functionally interacting with the MCAT, KS I, and reductase domains from the *E. coli* FAS when heterologously expressed (Tropf et al (1998), PubMed ID: 9545424), we consider Rat ACP as an analog of the *E. coli* ACP with an inhibited sequestration capability. Therefore, the experiment represented by Fig 2A in the main text directly address this comment. We have revised the main text (lines 91-96) to more clearly explain the role of this experimental work.*

Trying QM/MM is also interesting to simulate the Raman shift of alkyne under sequestration. We completely agree with the Reviewer that this is an important in interesting step. The Londergan group (in collaboration with experts in two kinds of frequency simulation methodologies) showed in 2018 that state-of-the art QM/MM calculations of vibrational probe group signals do not provide excellent agreement with experimental data, an effective fragment potential approach (introduced originally by Blasiak and Cho, and reviewed in reference 13) yields both quantitative agreement with experiment and clear physical interpretation of changes in probe group frequencies and lineshapes. This methodology has unfortunately not yet been extended to alkynes, whose solvatochromism is a relatively newly observed phenomenon. Such

an extension is well beyond the scope of the current work, but we are currently undertaking this computational work and expect to be able to directly compare simulations to experimental data like that reported here in the near future. We note this important information in two new sentences in the manuscript, on lines 172-175.

Reviewer #2 (Remarks to the Author):

This is a valuable contribution to the acyl carrier protein field. The sample preparations and quality control are exemplary. The data are solid. We thank the Reviewer for this positive feedback!

*However, the authors state that these are facile and quick measurements. That seems true if one has the proper Raman spectrometer. The authors used a home-built instrument. Are there commercial instruments of similar abilities? If so, how likely are they to be accessible to biochemists? Most Raman spectrometers I see are in material sciences labs. The authors need to address this aspect. This is an excellent point. The spectra collected here use a relatively unsophisticated continuous-wave Raman spectrometer, with no optical enhancement techniques (i.e. UV resonance, stimulated Raman, surface or plasmonic enhancement, etc.). While commercial FT-Raman spectrometers might not have the signal:noise level to easily collect these signals, most Raman spectrometers (including commercial dispersive Raman microscopes) should be capable of viewing these signals from similar *in vitro* samples. The Raman imaging techniques cited towards the end of the manuscript (refs. 24-26) use stimulated Raman to collect similar signals at much lower concentrations than we have in the samples reported here. So while Raman spectrometers are not standard equipment in many labs and many are somewhat different from each other, these signals are easy to collect and see using relatively nonspecialized equipment. We now comment on this briefly in the revised text on lines 166-169.*

Reviewer #3 (Remarks to the Author):

In this short, well written report, the authors introduce a new method for assessing the solvent exposure of acyl substrate mimics that have been covalently attached to the 4'-phosphopantetheine (Ppant) arms of acyl carrier protein (ACP) domains. The approach constitutes an ingenious extension of previous techniques devised by the corresponding author, who first followed changes in the ¹⁹F NMR signal of a trifluoromethyl group appended to the Ppant prosthetic group and then monitored changes in the vibrational signature of a similarly attached thiocyanate probe using infrared absorption spectroscopy. The current advance is to use Raman spectroscopy to detect changes in the signature of a carbon-carbon triple bond incorporated into Ppant-attached acyl substrate mimics of various lengths. The method can indicate whether the alkyne group experiences a non-polar or an aqueous environment in the absence of more detailed but time-consuming structural studies by X-ray crystallography or NMR spectroscopy. It will therefore be useful for testing the idea that type II fatty acid synthase (FAS) and polyketide synthase ACP domains sequester their substrate into a hydrophobic cavity, while type I ACPs leave their substrates solvent-exposed. Overall, in my opinion the article is of wide interest and is worthy of publication in this journal. We appreciate the thoughtful summary of our work and the positive feedback.

However, I also feel that the authors need to address the following points:

(1) If the alkyne substrate-mimic appears to remain solvent-exposed when attached to an ACP domain, an alternative explanation is that the protein might not be properly folded. The authors do not rule out this possibility in their report. From one of the corresponding author's recent publications (Rivas et al, 2018), it seems that circular dichroism data has been collected for all of

the domains studied except for the rat FAS ACP. Such information indicates that at least the expected alpha-helical secondary structure is present, so it should be referred to here, perhaps as a final validation step in the “Protein expression and purification” section of the Methods. Confirmation that the rat FAS ACP is structured should also be presented. The phosphopantetheine transferase Sfp is promiscuous enough to modify short unstructured peptides that possess the correct recognition sequence (e.g. see Yin et al (2005) PNAS 102, 15815-15820); successful modification by this enzyme (as observed here for all cases) is therefore not sufficient to prove that an ACP domain possesses native tertiary structure. We thank the Reviewer for this thoughtful suggestion. As the Reviewer notes, the CD spectra for all the *holo*-ACPs used in this study, except for that of Rat ACP, have previously been reported in Rivas et al (2018). To respond to this comment, we conducted additional experiments (Figure S8) reporting the CD spectrum for *holo*-Rat ACP and the spectra of probe-loaded ACPs: EcACP (with C5, C8, and C13 probes) and Rat ACP (with C8 probe). The new data indicates that the alpha helical secondary structure of the ACPs are not compromised by the addition of alkyne-containing substrates to the ACP. The EcACP and Rat ACP are notably representative of both sequestration states: sequestered with the hydrophobic cavity and solvent exposed. Additional information about the methodology behind these experiments has been added to the Methods section (lines 373-381). We feel as though the addition of these CD experiments increase the rigor of our manuscript and are grateful to the Reviewer for this suggestion.

(2) In addition to full burial and full exposure of the acyl chain, a third option is that substrate-mimics may nestle against a non-polar patch on the surface of the ACP domain (e.g. see Vance et al (2016) Biochem J 473, 1097-1110). It is not obvious how this mode of interaction would affect the vibrational frequency of the alkyne group, so it might be appropriate for the authors to include it in their suggestions for future work. This is an excellent point. As the alkyne-based probing method we report is amendable to studying a wide range of acyl carrier proteins, the implementation of our method into systems like those reported of the mycolactone ACP (Vance et al (2016)) that experience unconventional sequestration behavior would likely provide noteworthy results and start to elucidate these protein behaviors. In unpublished work, we have observed the thiocyanate probe displays a small (~ 2 cm⁻¹) shift with a narrow bandwidth when nestled against a patch on the surface of a protein, and we expect to see a similar trend for the alkyne probe. We modified our text related to future directions to include the possibility of visualizing unconventional behaviors such as the one noted by the Reviewer as a future direction (lines 184-186).

(3) All of the substrate-mimics tested in the report possess chain-terminating alkyne groups. It would be interesting to learn how the exposure of this group changes if it is moved further up the chain. Perhaps this could also be included in the list of future applications. Yes, we are excited about this possibility as well! We have now included text reflecting on this possible future direction (lines 186-189).

(4) In line 143, the WhiE ACP is not introduced adequately. It also appears that this construct not fully described elsewhere in the text, or in the prior publication Rivas et al (2018). I imagine that the authors intend this to indicate the product of the WhiE gene from the grey spore pigment biosynthetic cluster in Steptomyces strain. The source of this construct should be described in the text more fully. We appreciate the Reviewer bringing this important point to our attention. We have modified the text to reflect that the WhiE ACP belongs to the spore pigment biosynthetic gene cluster (see lines 149-150).

(5) In line 111, the claim to have achieved “unprecedented temporal resolution” requires some moderation. The authors explain convincingly why their method ought to be capable of achieving

temporal resolution, but the experiments presented here do not show this. We appreciate this thoughtful and candid feedback. Our claim of “unprecedented time resolution” is based on the intrinsic time scale of the Raman spectroscopy measurement and its ability to distinguish spectral populations on time scales that are many orders of magnitude faster than NMR spectroscopy. The intrinsic time scale, or “frame rate” of the vibrational spectrum in this frequency range is picoseconds at its absolute slowest; the time scale at which signals like those observed here would fall in the “fast exchange” limit is approximately 1 ps. This can be demonstrated in several brief calculations:

- One period of the alkyne CC stretching mode (approximately 2100 cm^{-1}) occurs in about 16 fs
- The frequency difference between sequestered and non-sequestered alkyne peaks (a difference of about 5 cm^{-1}) corresponds to a time delay (reciprocal of the frequency difference) of about 7 ps: this is roughly the exchange lifetime of the two subpopulations below which the peaks would be observed as partially coalesced (rather than separate spectral subpopulations)
- A lineshape simulation that assumes two Gaussian spectral subpopulations at 2105 and 2110 cm^{-1} , a vibrational lifetime of about 5 ps (a reasonable assumption that comes from nonlinear vibrational experiments performed elsewhere) and includes variable exchange rates between the two populations (now included as Figure S9) demonstrates that substantial narrowing/coalescence of the two spectral subpopulations can only occur when the spectral subpopulations interconvert on the time scale of 10 ps or faster.

The 10s of picoseconds time scale is substantially slower than all anticipated Ppant arm motions. MD simulations indicate that sequestration could proceed on the ns or even several 100s of ps time scales, but not faster: this means that we expect all Ppant arm configurations, as probed by an alkyne on the end of the arm’s substrate, to be in slow exchange in experiments like those reported here in Fig. 2 (even if the spectral subpopulations are separated by the resolution of the spectral measurement, which here is about 2 cm^{-1}). In response to this comment, we have added a specific mention of this time scale to the revised manuscript on line 116-118, and we have included the spectral simulation mentioned here in the supporting information as Figure S9.

(6) In line 42, the modifiable serine residue is described as being “at the end of CP helix II”. It would be more instructive to change this statement to “at the N-terminal end of CP helix II”. We agree and have modified the text accordingly (line 42).

Sincerely,

Lou Charkoudian
Assistant Professor, Chemistry

REVIEWERS' COMMENTS:

Reviewer #1 (Remarks to the Author):

The authors properly addressed the issues raised by the reviewers. It is recommended to publish this manuscript in the current form.

Reviewer #2 (Remarks to the Author):

My concerns have been satisfied

Reviewer #3 (Remarks to the Author):

The authors have addressed all of the reviewers comments satisfactorily in their response and in the revised version of the manuscript. I recommend that they make for following minor corrections prior to publication:

- (1) line 19: change "visualization the" to "visualization of the"
- (2) line 38: change "methods in protein structure" to "protein structure methods"
- (3) line 118: change "can be" to "should be"
- (4) line 164-165: change "method will enable" to "method that will enable"
- (5) line 168: too much use of the word "here", so perhaps change "reported here" to "we report"
- (6) line 169: change "present here of" to "present of"
- (7) line 173: change "those" to "such"
- (8) line 174: change "and should" to "which should"
- (9) line 175: omit "data"

REVIEWERS' COMMENTS:

Editorial Request	Author Response
Reviewer #1 (Remarks to the Author): The authors properly addressed the issues raised by the reviewers. It is recommended to publish this manuscript in the current form.	n/a
Reviewer #2 (Remarks to the Author): My concerns have been satisfied	n/a
Reviewer #3 (Remarks to the Author): The authors have addressed all of the reviewers comments satisfactorily in their response and in the revised version of the manuscript. I recommend that they make for following minor corrections prior to publication: (1) line 19: change "visualization the" to "visualization of the" (2) line 38: change "methods in protein structure" to "protein structure methods" (3) line 118: change "can be" to "should be" (4) line 164-165: change "method will enable" to "method that will enable" (5) line 168: too much use of the word "here", so perhaps change "reported here" to "we report" (6) line 169: change "present here of" to "present of" (7) line 173: change "those" to "such" (8) line 174: change "and should" to "which should" (9) line 175: omit "data"	All comments have been addressed accordingly